# COMEM: Context Management with A Decoupled Long-Context Model

**Yuwei Zhang** [1]  **Chengyu Dong** [2]  **Shuowei Jin** [2]  **Changlong Yu** [2]  **Hejie Cui** [2]  **Hongye Jin** [2]  **Xinyang Zhang** [2]
**Hamed Bonab** [2]  **Colin Lockard** [2]  **Jianshu Chen** [2]  **Zhenyu Shi** [2]  **Jingbo Shang** [1]  **Xian Li** [2]  **Bing Yin** [2]

## Abstract

Context management enables agentic models to solve long-horizon tasks through iterative summarization of previous interaction histories. However, this process typically incurs substantial decoding overhead for the extra summarization tokens, which significantly affect the end-to-end response latency at deployment. In this paper, we introduce COMEM, a novel framework that decouples memory management from the primary agent workflow, enabling these processes to execute in parallel. We propose a $k$-step-off asynchronous pipeline that overlaps the memory model's summarization with the agent's inference, effectively masking the latency of context processing. To ensure robustness under this asynchronous setting, we introduce a reward-driven training strategy that aligns the memory model to capture sufficient statistics for the agent's decision-making. Theoretical analysis confirms that COMEM offers a superior efficiency-effectiveness trade-off compared to coupled architectures. Our extensive experimental results on SWE-Bench-Verified show that COMEM provides 1.4x latency improvements upon vanilla long-context solutions while preserving most of the performance. Furthermore, we demonstrate that these latency gains scale favorably with increased system throughput, offering a modular path forward for the independent optimization of agent reasoning and memory compression.

## 1. Introduction

The capabilities of Large Language Model (LLM) agents have expanded significantly, moving beyond simple conversational assistants to autonomous systems capable of tackling complex, multi-step problems in domains such as software engineering (Jimenez et al., 2024), scientific discovery (Lu et al., 2024), and open-ended exploration (He et al., 2024; Team et al., 2025). Unlike standard question-answering tasks, the success on these applications depends not only on the immediate instruction but on the agent's ability to maintain a coherent understanding of its entire interaction history. For instance, in repository-level code generation, an agent must recall decisions made thousands of steps prior to generate valid subsequent actions. Consequently, the ability to process and reason over extremely long contexts has become a non-negotiable requirement for high-performing agentic systems (Zhou et al., 2025; Wu et al., 2025b; Lu et al., 2025; Sun et al., 2025; Yu et al., 2025).

Despite the increasing attention to extend the effective context window, there has been surprisingly little attention paid to the severe computational costs introduced by long-context processing during agentic inference. While modern LLMs can natively accept context windows spanning millions of tokens, utilizing them during online agent deployment is often prohibitively expensive (Yuan et al., 2024). The primary challenge lies in the inference latency of the decoding stage. As the interaction history grows, the KV cache footprint expands linearly, and the attention computation cost scales, saturating the High Bandwidth Memory (HBM) of modern GPUs. Unlike the prefilling phase, which is compute-bound and parallelizable, the auto-regressive decoding phase is memory-bound, since fetching the entire KV cache for every generated token results in high latency that degrades the user experience and limits the throughput of real-time systems (Tang et al., 2024).

To address these computational challenges, previous solutions have largely pursued two directions: context reduction and system-level optimization. Context reduction strategies attempt to limit the number of tokens the model must attend to. Sliding-window attention (Fu et al., 2025) restricts the agent to a fixed local context, inherently sacrificing the ability to recall distant dependencies, such as the initial user goal or a bug identified thousands of steps prior. Retrieval-Augmented Generation (RAG) (Wu et al., 2025a; Shi et al., 2026) alleviates this by fetching relevant snippets from the history. However, RAG relies heavily on semantic similarity,

---

[*]Equal contribution [1]Halıcıoğlu Data Science Institute, University of California, San Diego [2]Amazon. Correspondence to: Yuwei Zhang <yuz163@ucsd.edu>.

which often fails to capture the high-level state or procedural trajectory of an agent, leading to fragmented reasoning.

On the other hand, system-level optimizations aim to make processing the full context more efficient without reducing the number of tokens. Modern inference engines like vLLM (Kwon et al., 2023) have revolutionized memory management through PagedAttention, significantly reducing fragmentation. Similarly, Sparse Attention mechanisms (Xiao et al., 2024) and KV Cache Quantization (Hooper et al., 2024) reduce the compute and memory bandwidth requirements by sparsifying interactions or lowering numerical precision. While these methods provide substantial throughput improvements, they do not solve the the redundancy of re-encoding or re-attending to a massive, largely static interaction history for every single decoding step. As a result, even highly optimized engines eventually hit a latency wall as the interaction trace grows indefinitely.

To bridge this gap, we introduce CoMem, a framework that decouples the distinct responsibilities of memory management and agentic reasoning. Unlike prior approaches that force a single large model to handle both history compression and policy generation, CoMem offloads the heavy lifting of long-context processing to a dedicated, lightweight summarization model. This architectural separation allows us to propose a novel $k$-step-off asynchronous pipeline, where the memory model continuously compresses history in the background, freeing the main agent to decode with a significantly reduced context window. To ensure this compressed state effectively guides the agent, we introduce a reward-driven alignment strategy that trains the memory model to capture the "sufficient statistics" required for optimal decision-making. By shifting the computational burden from the critical path of decoding to a parallelizable background process, CoMem achieves the best of both worlds: the global reasoning capability of long-context models and the low-latency inference of short-context systems.

Our contributions are summarized as follows:

- We propose CoMem, a framework that separates memory management from reasoning, enabling the use of specialized, lightweight models for efficient history compression.

- Asynchronous Inference: We design a $k$-step-off pipeline that overlaps memory summarization with agent execution, masking the latency of context processing and reducing the decoding overhead by up to $1.4\times$.

- We introduce a novel training methodology that aligns the memory model using a functional equivalence reward, ensuring the compressed summary recovers the full-context agent's policy without expensive online interaction.

- Extensive experiments on SWE-Bench-Verified demonstrate that CoMem significantly reduces latency while maintaining competitive performance against state-of-the-art long-context baselines. Code is available at: https://github.com/horizon-llm/CoMem.git.

## 2. Preliminaries

### 2.1. Standard Agent

Standard agent interaction is a *Partially Observable Markov Decision Process* (POMDP), where each process starts from an initial observation $o_1$ that usually contains a problem statement, and then a multi-turn interaction history of length $T$ is generated as

$$\tau = (o_1, a_1, o_2, a_2, \ldots, o_T)$$

where $a_i$ usually contains a reasoning block and a tool call, and $o_i$ usually represents an observation returned from the environment. The standard Large Language Model (LLM) agent takes the entire interaction history $\tau$ as input at each step to infer the internal belief state of the environment, and generates the next action $a_i$. This results in a linearly scaling context length during the interaction that pressures the model's inference efficiency and performance.

### 2.2. LLM Inference

Modern LLMs usually employ a two-stage inference process: **Prefill Stage** and **Decode Stage**. The Prefill Stage serves as the initial step where the key-value cache (KV cache) for the prompt (also called prefix) sequences are built for each KV head in the Transformer architecture[1]. Given hidden states $\mathbf{X} \in \mathbb{R}^{l \times d_{\mathrm{model}}}$, each attention layer computes $\mathbf{Q} = \mathbf{X}\mathbf{W}_q \in \mathbb{R}^{l \times h_q \times d_h}$, $\mathbf{K} = \mathbf{X}\mathbf{W}_k \in \mathbb{R}^{l \times h_{\mathrm{kv}} \times d_h}$, and $\mathbf{V} = \mathbf{X}\mathbf{W}_v \in \mathbb{R}^{l \times h_{\mathrm{kv}} \times d_h}$ after reshaping, where $h_q$ is the number of query heads, $h_{\mathrm{kv}}$ is the number of KV heads, and $d_h$ is the head dimension. After prefilling, all the key-value pairs are preserved in the memory so that they can be reused during the decoding of each token. This will result in two vectors $\mathbf{K}_{\mathrm{cache}} = [\mathbf{K}_1; \ldots; \mathbf{K}_l] \in \mathbb{R}^{l \times h_{\mathrm{kv}} \times d_h}$ and $\mathbf{V}_{\mathrm{cache}} = [\mathbf{V}_1; \ldots; \mathbf{V}_l] \in \mathbb{R}^{l \times h_{\mathrm{kv}} \times d_h}$. The GPU High Bandwidth Memory (HBM) typically needs to spare a significant amount of spaces to store these KV cache. For instance, for `Qwen/Qwen3-32B` in FP16 precision, it will take up $16$ GB to save a sequence of $64$K tokens. In contrast, `Qwen/Qwen3-4B` requires roughly half of the space to save these amount of KV cache[2]. The Prefill Stage is usually *compute-bound*, since it involves matrix-matrix mul-

---

[1]Note that here we discriminate KV head from attention head because of Grouped Query Attention (GQA)

[2]https://lmcache.ai/kv_cache_calculator.html provides calculations for some other models

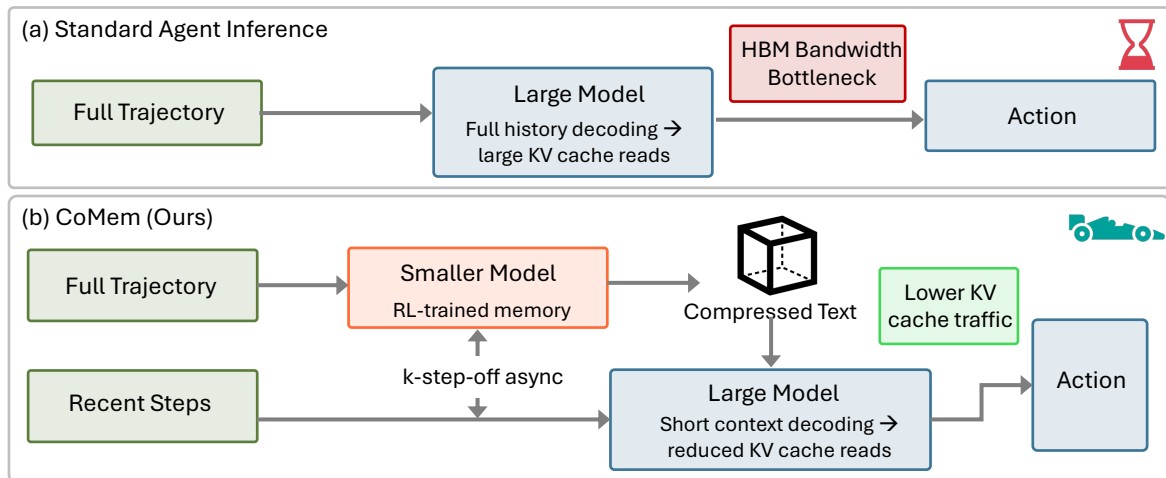

*Figure 1.* CoMem framework: a decoupled agent framework that offloads long-context compression to an asynchronous, lightweight memory model, significantly reducing inference latency without compromising reasoning performance.

tiplications that saturates the GPU's arithmetic units. We use *Time to First Token* (TTFT) to calculate the time spent from waiting the first token to emit to represent the prefilling latency.

Once the prompt is processed, the model enters the Decode Stage. For each step $t$, the model processes the current prefix and concatenates a new KV pair: $\mathbf{K}_{\text{cache}}^{(t)} = [\mathbf{K}_{\text{cache}}^{(t-1)}; \mathbf{K}^{(t)}]$, $\mathbf{V}_{\text{cache}}^{(t)} = [\mathbf{V}_{\text{cache}}^{(t-1)}; \mathbf{V}^{(t)}]$. And at the same time emits the output token. Different from Prefill Stage, Decode Stage is memory-bound since at each decoding step, the GPU spends more time waiting for the KV cache to arrive from HBM. Consequently, the decoding speed is usually bottlenecked by the memory bandwidth. When the HBM is saturated, part of the KV cache will be either emitted or offloaded to CPU (Jin et al., 2025). CPU offload is often a better strategy which results in linear prefilling costs compared to the quadratic compute in long-context prefilling. Typically, we measure decoding efficiency through *Time per Output Token* (TPOT), that tracks the average generation speed per token, excluding the initial prefilling costs.

### 2.3. Group Relative Policy Optimization (GRPO)

In Group Relative Policy Optimization (GRPO) (Shao et al., 2024), for each prompt $x$, we sample a group of $K$ trajectories $\mathcal{G}(x) = \{\tau^{(1)}, \dots, \tau^{(K)}\}$ where each trajectory $\tau^{(k)}$ is generated by the policy $\pi_\theta$ and receives a trajectory-level scalar return $R(\tau^{(k)})$. GRPO constructs a *relative* advantage within each group by subtracting a group-wise baseline from the individual return. Concretely, we define $\tilde{A}^{(k)} = R(\tau^{(k)}) - b(\mathcal{G}(x))$ where $b(\mathcal{G}(x))$ is a baseline that depends only on the group statistics: $b(\mathcal{G}(x)) = \frac{1}{K} \sum_{j=1}^{K} R(\tau^{(j)})$. The resulting policy gradi-

ent objective over a batch of groups is

$$J(\theta) = \mathbb{E}_{x, \mathcal{G}(x)} \left[ \frac{1}{K} \sum_{k=1}^{K} \tilde{A}^{(k)} \sum_{t} \log \pi_\theta \big( a_t^{(k)} \mid s_t^{(k)} \big) \right],$$

where the inner sum runs over all decision tokens in the trajectory. In practice, we use a PPO-style clipped surrogate to stabilize training. Let $r_t^{(k)} = \frac{\pi_\theta(a_t^{(k)} | s_t^{(k)})}{\pi_{\theta_{\text{old}}}(a_t^{(k)} | s_t^{(k)})}$ denote the likelihood ratio; the GRPO loss is

$$\mathcal{L}_{\text{GRPO}}(\theta) = -\mathbb{E} \left[ \frac{1}{K} \sum_{k,t} \min \left( r_t^{(k)} \tilde{A}^{(k)}, \right. \right.$$
$$\left. \left. \text{clip} \left( r_t^{(k)}, 1 - \epsilon, 1 + \epsilon \right) \tilde{A}^{(k)} \right) \right] \tag{1}$$
$$+ \beta \, \mathbb{E} \left[ D_{\text{KL}} \big( \pi_\theta \, \| \, \pi_{\text{ref}} \big) \right]$$

where $\epsilon$ controls the clipping range, $\beta$ weights a KL penalty to a reference policy $\pi_{\text{ref}}$, and the expectation is taken over prompts, groups, and time steps.

## 3. Design of CoMem

We design CoMem to optimize the latency during high-throughput agentic inference. By decoupling the long-context processing from agentic process and then overlapping them via $k$-step-off generation, our method effectively alleviates the memory-bound bottlenecks and improves the end-to-end performance while keeping most of the accuracy. In the following sections, we first analyze the decoding latency bottleneck in agentic inference and show that KV cache loading dominates under high-throughput settings (§3.1). Next, we present the key insights of our proposed framework CoMem which includes a novel $k$-step-off inference strategy (§3.2), and a new RL training pipeline that

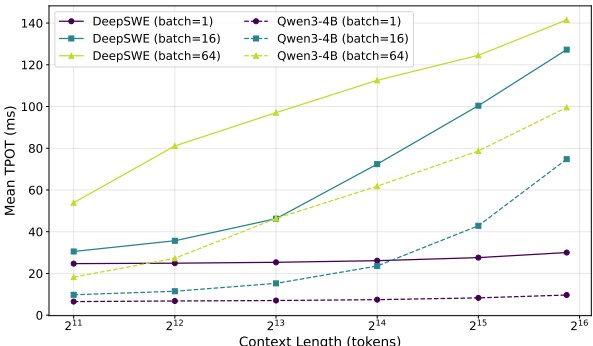

*Figure 2.* TPOT for two models on various context lengths and batch sizes. DeepSWE is a dense model with 32B parameters.

encourages the model to capture sufficient statistics (§3.4). Finally, we show the theoretical analysis on the determination of a proper summary length (§3.3).

### 3.1. Analysis of Agent Inference

In agentic inference, since at each step, only the current observations are prefilled, the total time is typically dominated by the decoding stage, which is often memory-bound because of the loading of KV cache. Thus, we study the decoding speed in this section in both synthetic long-context and standard agent inference scenarios.

**Long-context Scenario.** Real-time agentic inference often involves additional overheads from external processes such as I/O, as well as varying batch sizes and token lengths during rollout. To isolate the relationship between context length and decoding time, we first conduct a profiling analysis on a synthetic long-context scenario in which inputs and outputs are random data with fixed lengths and a constant batch size. As shown in Figure 2, when the batch size is small (batch=1), TPOT does not scale significantly with context length, because the time spent loading the KV cache is negligible compared with other overheads such as loading model weights. For batch=16, at shorter contexts ($< 2^{13}$), the KV cache remains small enough that GPU memory bandwidth is not saturated, and TPOT is still dominated by fixed overheads. Beyond $2^{13}$ tokens, however, the KV cache becomes large enough that each additional token incurs a linear increase in latency. For large batch sizes (batch=64), the GPU memory bandwidth is fully saturated even at short contexts, so TPOT scales linearly throughout.

**Agentic Scenario.** To further investigate how context-length-induced slow-down manifests during agentic inference, we measure the per-step execution time and its constituent components, averaged across multiple trajectories generated on SWE-Bench-Verified. The results are presented in the left panel of Figure 3. We make two key observations. First, the total wall-clock time at each step is predominantly attributed to LLM execution, whereas envi-

ronment execution constitutes a negligible fraction. Second, the LLM execution time exhibits a non-monotonic trend, increasing progressively from the initial steps up to approximately step 20, after which it begins to decrease toward the final steps. We attribute this non-monotonic behavior to the evolving batch-size dynamics during inference. In the early stages, the large batch size renders the decoding process memory-bound, leading to increased latency as context lengths grow. As the generation progresses and individual requests reach completion, the effective batch size diminishes, thereby alleviating the memory bandwidth bottleneck and reducing per-step inference latency. To disentangle the effect of prompt length from that of batch-size variation, we further present the execution time per completion token as a function of prompt length in the right panel of Figure 3 (orange line). The results demonstrate that decoding latency per token increases monotonically with prompt length, thereby corroborating the hypothesis that longer contexts incur proportionally greater decoding overhead.

### 3.2. CoMem Framework

Through the preceding analysis, we demonstrate that inference latency is primarily bottlenecked by the decoding stage. While most agentic models are large reasoning models with substantial decoding overhead, there exist several lightweight long-context models capable of offloading the burden of long-context processing from the decoding phase. In this section, we introduce CoMem (Figure 1), that decouples memory management from the main agent workflow by employing a dedicated smaller memory model that compresses prior interactions into a concise representation for the agent. This design reduces latency and memory footprint since fewer parameters are involved, while effectively maintaining a long-context processing capability. Specifically, (1) **memory model** ($f$) is a native long-context model with a lightweight decoding mechanism (we choose a model with smaller parameter count) and is designed to capture sufficient statistics from long-term history while (2) **agent model** ($\pi$) is a more capable model (usually with larger parameter counts) that proposes policies based on the outputs from $f$ and the short-term memory.

Formally, the memory model $f$ maps the long-term history to a compressed state representation $s_t$:

$$s_t = f(\tau_{<t}; \theta_f) \tag{2}$$

where $|\theta_f| \ll |\theta_\pi|$, ensuring that the computational burden of long-context encoding is handled by a smaller, more efficient model. Moreover, the constraint $|s_t| < |\tau_t|$ guarantees that the input length for the agent is significantly reduced, resulting in lower decoding latency compared to standard full-context inference. Then the agent model proposes pol-

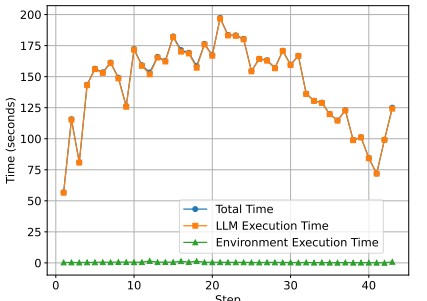 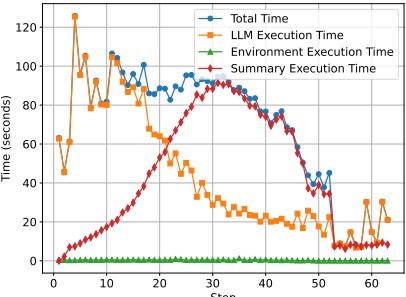 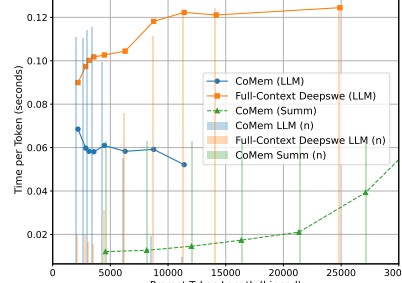

*Figure 3.* Profiling results for agentic inference scenario. Left is the execution time for Full-Context baseline. Middle is the execution time for COMEM. Right is the time per completion token results.

icy based on the compressed summary.

$$a_t \sim \pi(a|s_t, C_t; \theta_\pi) \qquad (3)$$

A naïve implementation of the workflow described above introduces a *sequential bottleneck*, as the agent model must idle until the memory update concludes. Next, we introduce a novel pipeline to remedy this extra overhead via overlapping the generation of two models.

**$k$-step-off Pipeline** To mitigate this sequential bottleneck, we propose a novel $k$-step-off asynchronous pipeline (Algorithm 1), illustrated in Figure 4. The core idea is to overlap memory compression with agent execution by allowing the memory model to lag $k$ steps behind. The pipeline proceeds in cycles of $k$ steps. At the beginning of each cycle, the memory model is launched asynchronously to compress all history up to the current step. While the memory model runs in the background, the agent continues to execute for the next $k$ steps using the most recently available summary $s$ concatenated with a buffer of raw recent interactions. Upon completion, the new summary becomes available at the start of the next cycle, at which point the agent's KV cache is invalidated and a full uncached prefill is required. For the remaining $k - 1$ steps in the cycle, the agent reuses its cached prefix and performs only incremental prefilling of new observations which incurs negligible overhead compared to full-context inference. Formally, the policy generation at step $t$ is defined as:

$$a_t \sim \pi(a_t \mid \underbrace{f(\tau_{\leq t-k})}_{\text{Latent Summary}}, \underbrace{\tau_{t-k+1:t}}_{\text{Recent Buffer}} ; \theta_\pi) \qquad (4)$$

This design ensures that while the heavy lifting of long-context compression happens asynchronously, the agent always has access to the full context via both summary and the explicit short-term buffer. Larger $k$ amortizes the uncached prefilling cost over more steps (as shown in Figure 4, comparing 1-step-off with 2-step-off), at the expense of increased staleness in the summary. We will formalize the latency–staleness trade-off in the next section.

**Algorithm 1** $k$-step-off Pipeline
***
**Require:** Agent Model $\pi$, Memory Model $f$, Retain Turns $k$
1: **Initialize:** $s \leftarrow \emptyset$, $t \leftarrow 1$, $\mathcal{P}_{mem} \leftarrow$ None
2: **while** not finished **do**
3:     {**Synchronization (every $k$ steps)**}
4:     **if** $\mathcal{P}_{mem}$ is complete **then**
5:         $s \leftarrow$ result($\mathcal{P}_{mem}$);   invalidate KV cache
6:         $\mathcal{P}_{mem} \leftarrow$ None
7:     **end if**
8:     {**Agent Inference**}
9:     **if** KV cache valid **then**
10:        Prefill new tokens only: $P^\pi(o_t)$ {Cached}
11:     **else**
12:        Prefill full context: $P^\pi(s, \tau_{[t-k+1:t]})$ {Uncached}
13:     **end if**
14:     Decode $a_t \sim \pi(a \mid s, \ \tau_{[t-k+1:t]})$
15:     {**Environment Execution**}
16:     Execute $a_t$, observe $o_{t+1}$
17:     $\tau_{t+1} \leftarrow \tau_t \cup \{a_t, o_{t+1}\}$
18:     {**Launch Memory (async, every $k$ steps)**}
19:     **if** $t \bmod k = 0$ **then**
20:        $\mathcal{P}_{mem} \leftarrow$ async $f(\tau_{\leq t})$
21:     **end if**
22:     $t \leftarrow t + 1$
23: **end while**
***

### 3.3. Determination of Summary Length

While COMEM reduces memory overhead during the decoding stage, the introduction of the summary representation $s$ requires an additional prefilling stage for the agent model before further decoding, as shown in Figure 4. To ensure a net reduction in end-to-end latency, the decoding speedup must sufficiently amortize the prefilling cost. We formalize this trade-off below.

**Theoretical Analysis** Let $S$ denote the KV cache memory size required per token (in GB), and let $W$ represent the GPU's HBM bandwidth (in GB/s). Assuming the decoding process is memory-bound, the per-token latency reduction achieved by COMEM relative to the full-context baseline is:

$$\Delta t_{\text{decode}} = \frac{(L_{\text{full}} - L_{\text{sum}}) \cdot S}{W} \qquad (5)$$

where $L_{\text{full}}$ and $L_{\text{sum}}$ are the context lengths of the baseline and COMEM, respectively. However, generating the

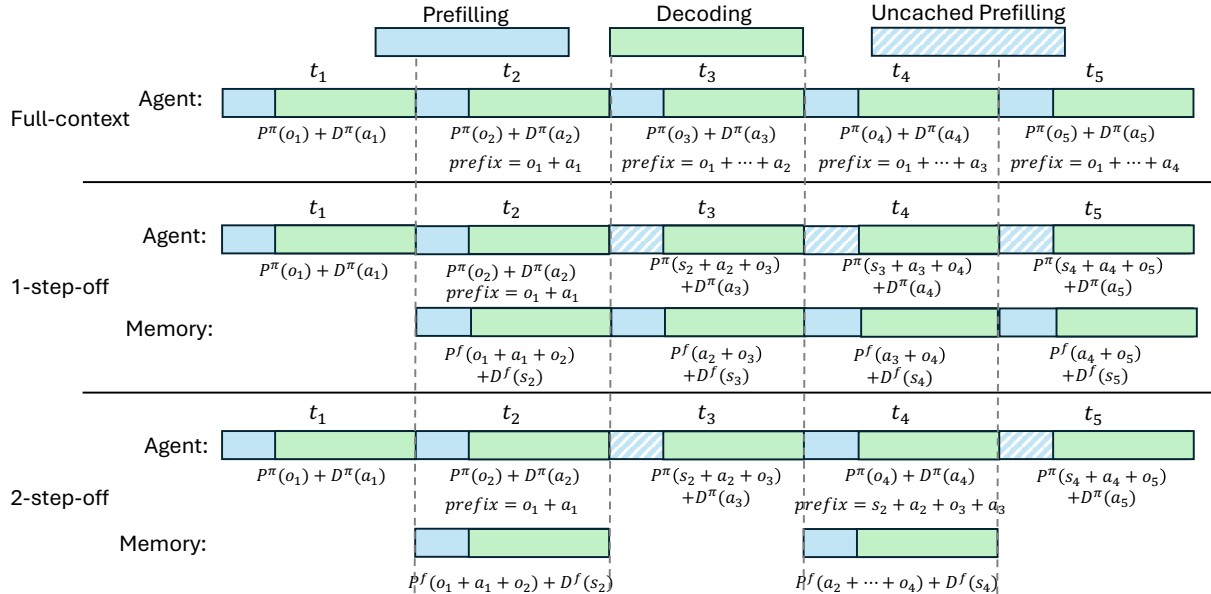

Figure 4. Gantt chart illustrating the $k$-step-off pipeline. We denote $P^\pi(\cdot)$ and $D^\pi(\cdot)$ as the prefilling and decoding time of the agent model, and $P^f(\cdot)$ and $D^f(\cdot)$ for the memory model. Solid blue blocks denote cached prefilling (incremental), hatched blocks denote uncached prefilling (full context re-encoding after a summary update), and green blocks denote decoding. The "prefix" annotations indicate the cached KV prefix carried over from the previous step. When the prefix remains valid, only new tokens require prefilling. For 1-step-off, every step after the first requires uncached prefilling due to the continuously updated summary. For 2-step-off, the uncached prefill occurs only once per cycle, with the intermediate step reusing the cached KV prefix.

summary representation incurs a one-time prefilling latency. Assuming a prefilling throughput of $P$ (tokens/s), a net speedup requires that the cumulative decoding savings over $Y$ generated tokens exceed the prefilling overhead:

$$Y \cdot \underbrace{\frac{(L_{\text{full}} - L_{\text{sum}}) \cdot S}{W}}_{\text{Per-token decoding saving}} > \underbrace{\frac{L_{\text{sum}}}{P}}_{\text{Prefill cost}} \qquad (6)$$

Rearranging yields an upper bound on the permissible compression ratio:

$$\frac{L_{\text{sum}}}{L_{\text{full}}} < \frac{1}{1 + \frac{W}{Y \cdot S \cdot P}} \qquad (7)$$

**Case Study.** We instantiate this bound for `Qwen3-32B` served on a single A100-80GB GPU, with HBM bandwidth $W = 2$ TB/s, prefilling throughput $P = 3,000$ tokens/s, per-token KV size $S = 2 \times 10^{-4}$ GB, and average completion length $Y = 1,000$ tokens. Substituting these values yields $L_{\text{sum}}/L_{\text{full}} < 0.23$, which means for a prefix length of $32K$ tokens, we need to upperbound the summary length within $7.36K$ tokens. This is a compression ratio readily achievable through summarization.

### 3.4. Training

Empirical analysis suggests that the inherent summarization capability of base models is often insufficient to compress the history for complex agentic interactions, especially with

a dedicated compression ratio. To bridge this gap, we introduce a training pipeline designed to align the summarization model's output with what the agent requires for optimal decision-making. The key idea is to train the memory model to capture the *sufficient statistics* of the interaction history—the minimal representation from which the agent can recover behavior equivalent to full-context inference (Sutton et al., 1998). Formally, a summary $s = f(\tau_{<t-k})$ is sufficient if the agent's action conditioned on the summary matches what it would produce given the full history: $\pi(\cdot \mid s, C_t) \approx \pi(\cdot \mid \tau_{<t})$, where $C_t = \tau_{t-k:t-1}$ denotes the most recent $k$ raw turns. Rather than optimizing for generic text quality, we directly optimize for this *functional equivalence*—whether the compressed memory induces correct downstream behavior.

Crucially, the agent model $\pi$ remains frozen throughout training. This decouples the optimization of the memory model $f$ from the agent's reasoning policy, enabling an efficient offline regime that requires no online environment interactions. Concretely, we first generate reference trajectories using the standard full-context agent pipeline. For each step $t$ in a trajectory, we record the ground-truth action $a_t^* = \pi(\cdot \mid \tau_{<t})$ produced by the full-context agent. We then fine-tune the memory model using GRPO (Eq. 1) with the following reward: given a candidate summary $s = f(\tau_{<t-k})$, the frozen agent produces $\hat{a}_t = \pi(\cdot \mid s, C_t)$, and we define

$$R(s) = \text{sim}(\hat{a}_t, \ a_t^*) \qquad (8)$$

*Table 1.* Main results. For latency, we show the total inference time for 128 issues.

| Agent | Memory | %Resolved | #Tool Calls | w/o CPU Offload | | w/ CPU Offload | |
|---|---|---|---|---|---|---|---|
| | | | | Latency ($\times128$/s) | Speedup | Latency ($\times128$/s) | Speedup |
| **DeepSWE** **(32B Dense)** | Full-Context | 40.4 | 34.39 | 9457.78 | $1\times$ | 6390.62 | $1\times$ |
| | Qwen3-4B (base) | 29.9 | 24.74 | 4360.72 | $2.17\times$ | 3649.61 | $1.75\times$ |
| | Qwen3-4B (SFT) | 39.8 | 35.61 | 5232.64 | $1.81\times$ | 4168.85 | $1.53\times$ |
| | **GRPO**$_{AC}$ | **41.0** | 40.69 | 5622.33 | $1.68\times$ | 4400.89 | $1.45\times$ |
| **Qwen3-Coder-Max** **(480B A35B)** | Full-Context | **57.2** | 36.68 | 6291.11 | $1\times$ | 5129.90 | $1\times$ |
| | No Summary | 46.6 | 25.20 | 3780.74 | $1.66\times$ | 3421.78 | $1.50\times$ |
| | Qwen3-4B (base) | 42.2 | 39.40 | 3421.63 | $1.84\times$ | 3138.97 | $1.63\times$ |
| | Qwen3-4B (SFT) | 47.3 | 43.39 | 3819.70 | $1.65\times$ | 3337.58 | $1.54\times$ |
| | **GRPO**$_{AC}$ | 51.0 | 44.55 | 3917.68 | $1.61\times$ | 3594.51 | $1.43\times$ |
| **GLM-4.7** **(355B 32B)** | Full-Context | **69.0** | 55.11 | 6361.49 | $1\times$ | 5869.42 | $1\times$ |
| | No Summary | 59.4 | 46.82 | 4216.83 | $1.51\times$ | 3842.17 | $1.53\times$ |
| | Qwen3-4B (base) | 58.3 | 49.32 | 3928.44 | $1.62\times$ | 3526.91 | $1.66\times$ |
| | Qwen3-4B (SFT) | 61.3 | 52.74 | 4087.92 | $1.56\times$ | 3668.53 | $1.60\times$ |
| | **GRPO**$_{AC}$ | 62.7 | 51.21 | 3318.42 | $1.92\times$ | 2821.38 | $2.08\times$ |

By maximizing this reward while enforcing a clipping on the maximum response length, the memory model learns to distill precisely the information the agent needs for decision-making, discarding irrelevant details that would otherwise consume context budget.

## 4. Experimental Results

### 4.1. Evaluation Dataset

We evaluate CoMem on SWE-Bench-Verified (Jimenez et al., 2024), a human-validated subset of the original SWE-bench dataset designed to ensure reliable and reproducible evaluation of autonomous software engineering agents. Specifically, following (Jain et al., 2025), we train our memory model using R2E-Gym-Lite and then test on 500 github issues on the SWE-Bench-Verified test set. We use the default scaffold provided by R2E-Gym and set maximum steps to 40 with an extended allowance to 100 steps to let the LLM submit the answer. Finally, we measure resolve rate, number of tool calls and inference latency for each batch of 128 issues.

### 4.2. Baselines and Implementations

We choose `Qwen3-4B` as our smaller memory model because of its scale and long-context capability. We always use the maximum summary length of 2048 for consistency. We pair the memory model with agent models with various sizes and capabilities. Specifically, we choose `DeepSWE`, `Qwen3-Coder-Max` and `GLM-4.7`. We compare with the following baselines: **Full-Context** is the standard agentic inference scenario with long-context inference. **No Summary** removes the summaries from the agent model prompt and leave the other components intact. **Qwen3-4B (base)** uses off-the-shelf model as memory model. Finally, **GRPO**$_{AC}$ is a model trained using our proposed framework

in Section 3.4. For `DeepSWE` and `Qwen3-Coder-Max`, we use $k = 2$, and for `GLM-4.7` we use $k = 4$. For latency comparison, we start two different vLLM engines for both agent model and memory model. We consistently choose vLLM version `0.14.0` with prefix caching and chunked prefilling. We further compare latency under two scenarios: with or without CPU offloading, to demonstrate the capability of CoMem on wide deployment settings. For `DeepSWE`, we run on A100 (80GB), and for the other two models, we run on H200. Notice that in production, one memory model server can be used for multiple agent models. See further discussions in Section 6. Detailed hyperparameters for both training and evaluation are provided in Appendix H.

### 4.3. Main Results

We evaluate CoMem on SWE-bench Verified across three agent backbones: `DeepSWE` (32B), `Qwen3-Coder-Max` (480B), and `GLM-4.7` (355B). In terms of effectiveness, our framework consistently outperforms standard compression baselines. Notably, on the `DeepSWE` backbone, CoMem (GRPO) achieves a 41.0% resolution rate, slightly surpassing the full-context baseline (40.4%), suggesting that aligned summarization can effectively filter irrelevant noise for mid-sized models. For larger models like `Qwen3-Coder-Max` and `GLM-4.7`, our reward-driven training significantly recovers performance compared to base and SFT variants, closing the gap with the full-context upper bound while restoring tool-use frequency. In terms of efficiency, the decoupled architecture delivers robust inference speedups ranging from $1.45\times$ to $2.08\times$ across different hardware configurations. The gains are most pronounced on GLM-4.7 (up to $2.08\times$), confirming that CoMem successfully mitigates the decoding bottleneck without compromising the agent's ability to solve complex, long-horizon tasks.

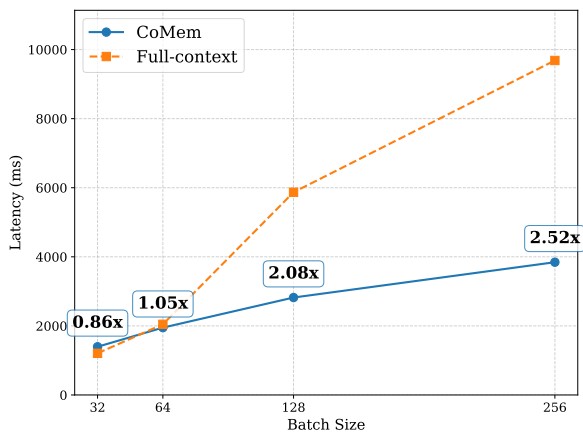

*Figure 5.* Latency and Speedup results for `GLM-4.7` over various batch sizes. COMEM's speedup scales up with throughput.

*Table 2.* Per-step execution time comparison across concurrency levels. Peak speedup denotes the maximum per-step speedup observed over the trajectory. "Conc." stands for concurrency.

| Conc. | COMEM (s) | FC (s) | Traj. Speedup | Peak Speedup |
|---|---|---|---|---|
| 16 | 28.4 | 28.9 | $1.02\times$ | $1.14\times$ (step 62) |
| 32 | 42.9 | 55.7 | $1.30\times$ | $2.03\times$ (step 56) |
| 64 | 52.3 | 88.8 | $1.70\times$ | $4.95\times$ (step 61) |

*Table 3.* KV cache utilization and prompt length comparison. FC prompt length is reported at the final step. "Conc." stands for concurrency.

| Conc. | COMEM KV | FC KV | COMEM Prompt | FC Prompt |
|---|---|---|---|---|
| 16 | 1–6% | up to 34% | 4.8–6.7K | 30.7K |
| 32 | 15–18% | up to 94% | 4.8–6.7K | 46.0K |
| 64 | 26–37% | up to 96% | 4.8–6.7K | 33.2K |

### 4.4. Scalable Speedup Gains with Increased Batch Size

We evaluate the inference latency of COMEM against a Full-context baseline across batch sizes ranging from 32 to 256. While the baseline performs competitively at the smallest scale, it exhibits poor scaling as batch size increases, with latency rising from 1206.60 ms to 9684.24 ms. In contrast, COMEM demonstrates superior computational efficiency and linear scaling; it achieves a $2.08\times$ speedup at a batch size of 128 and a $2.52\times$ speedup at 256, effectively mitigating the memory-bandwidth bottlenecks typical of large-batch long-context processing.

### 4.5. Crossover Analysis on Latency and Memory

To further provide practical intuition for when COMEM becomes advantageous, we conduct a controlled latency and memory benchmark comparing COMEM against the full-context baseline across varying serving concurrency levels (16, 32, and 64). The benchmark uses fixed 512 input tokens and 1024 output tokens over 64 steps, with `GLM-4.7` as the agent model and $k = 3$ for COMEM.

**Execution Time.** Table 2 reports per-step LLM execution time averaged over the trajectory. COMEM's per-step latency remains nearly constant ($\sim$28–52s depending on concurrency) because its prompt size is bounded at $\sim$5–7K tokens through periodic summarization. In contrast, the full-context baseline's prompt grows linearly with the number of steps, and its latency degrades accordingly.

**Memory Consumption.** Table 3 reports KV cache utilization and prompt lengths. COMEM's GPU KV cache usage remains bounded, while the full-context baseline's KV cache grows monotonically, reaching 96% at concurrency 64. At high concurrency, KV cache saturation triggers request preemption in the serving engine, causing the non-linear latency explosion observed in Table 2. The

crossover point where COMEM becomes faster than the full-context baseline depends on the serving regime: under concurrency 16, the two approaches are comparable as memory bandwidth is not yet saturated; under concurrency 32, COMEM becomes consistently faster once the prompt exceeds $\sim$25K tokens (around step 34); under concurrency 64, the crossover occurs at $\sim$12K tokens (around step 25), beyond which COMEM's advantage grows rapidly due to compounding KV cache pressure.

### 4.6. Sensitivity to $k$

In the $k$-step-off pipeline, the update frequency $k$ controls the trade-off between summary staleness and prefilling overhead: smaller $k$ provides fresher summaries but requires more frequent uncached prefills, while larger $k$ amortizes this cost at the expense of increased lag. We ablate this hyperparameter under CPU offload with the `GLM-4.7` agent backbone, reporting results in Table 4.

Performance is stable across $k \in \{2, 4, 8\}$, with $k = 2$ slightly outperforming the default $k = 4$ in resolve rate. The two extremes exhibit notably degraded behavior: $k = 1$ incurs excessive uncached prefilling overhead (as illustrated in Figure 4), while $k = 16$ saturates the memory of the agent server and introduces scheduling overhead. Importantly, all COMEM variants with $k \in \{2, 4, 8\}$ achieve approximately $2\times$ wall-clock speedup over the full-context baseline while retaining the majority of its resolve rate, confirming that the method is not overly sensitive to this choice within a reasonable operating range.

## 5. Related Works

Recent advancements in scaling large language models (LLMs) to long-horizon tasks have largely focused on overcoming fixed context window constraints through dynamic

*Table 4.* Sensitivity to update frequency $k$ on SWE-bench Verified with `GLM-4.7` (CPU offload enabled).

| Frequency $k$ | %Resolved | Latency (s) | Speedup |
|---|---|---|---|
| Full-Context | 69.0 | 5869.4 | $1.00\times$ |
| $k = 16$ | 60.2 | 3576.8 | $1.64\times$ |
| $k = 8$ | 62.4 | 2836.0 | $2.07\times$ |
| $k = 4$ (default) | 62.7 | 2821.4 | $2.08\times$ |
| $k = 2$ | 64.2 | 2841.4 | $2.07\times$ |
| $k = 1$ | 57.2 | 3843.5 | $1.53\times$ |

context management and memory optimization. One primary approach involves context compression and summarization, where methods like ReSum (Wu et al., 2025b), ACON (Kang et al., 2025), and SUPO (Lu et al., 2025) employ reinforcement learning to condense interaction histories into dense reasoning states, effectively discarding redundant information while retaining critical evidence. A parallel paradigm explores context folding and Markovian state reconstruction, illustrated by AgentFold (Ye et al., 2025), Context-Folding (Sun et al., 2025), and The Markovian Thinker (Aghajohari et al., 2026). These frameworks restructure linear history into branching or collapsible sub-trajectories, enforcing Markovian properties to decouple reasoning depth from input length. Finally, approaches such as Mem-$\alpha$ (Wang et al., 2025), MEM1 (Zhou et al., 2025), and DeepMiner (Tang et al., 2025) transition from passive context maintenance to active memory construction, training agents via RL to proactively update, retrieve, and synergize external memory stores with reasoning processes, thereby enabling sustained performance over indefinite horizons without linear computational scaling.

## 6. Discussion

**Practical Deployment and Model Synergy.** For real-world applications, COMEM enables a flexible and modular deployment strategy. We envision the memory model operating as a distinct high-throughput service, capable of handling context compression for multiple concurrent agent processes. This shared service architecture leverages the inherent efficiency of smaller models to offload context management, allowing for independent scaling of memory and reasoning components without necessitating specialized heterogeneous hardware.

To quantify this, we conduct an empirical cost analysis of the memory model server. From trajectory data, the memory model consumes $\sim$28K input tokens and $\sim$1K output tokens per compression call, with a mean step time of $\sim$40s, yielding a required throughput of approximately 156 tok/s per agent instance. Benchmarking the memory server under varying concurrency (1 to 128 concurrent requests), we observe that a 4B memory model achieves peak throughput of

$\sim$47K tok/s, supporting $\sim$300 concurrent `GLM-4.7` agent instances on a single server. This ratio scales gracefully with model size: an 8B model sustains $\sim$40K tok/s ($\sim$255 agents), and even a 32B model achieves $\sim$15K tok/s ($\sim$95 agents). In all cases, the memory model's serving cost is amortized across all agent instances sharing it, rendering COMEM's additional architectural complexity negligible at deployment scale.

**Relationship with Sparse Attention.** A prominent line of research for efficient long-context processing involves sparse attention mechanisms (Child et al., 2019; Beltagy et al., 2020; Zaheer et al., 2020), which mitigate the quadratic computational complexity of standard self-attention by limiting token interactions to predefined patterns or dynamic selections. More recent approaches, such as $H_2O$ (Zhang et al., 2023) and StreamingLLM (Xiao et al., 2023), further optimize inference by identifying "heavy hitter" tokens or utilizing attention sinks to prune the KV cache significantly. We emphasize that COMEM is orthogonal and complementary to these architectural optimizations. Sparse attention techniques operate at the *intra-model* level, optimizing how a single model attends to its context. In contrast, COMEM operates at the *system* level by decoupling the context management process entirely. Consequently, these approaches can be combined synergistically; for instance, the underlying Memory Model in COMEM could employ sparse attention or KV cache eviction policies to process raw interaction histories with even greater efficiency, while the Agent Model continues to benefit from the high-level, compressed state representations provided by our framework.

## 7. Conclusion

We proposed COMEM, a framework that decouples memory management from agentic reasoning by offloading long-context compression to a dedicated lightweight model via a novel $k$-step-off asynchronous pipeline. Our reward-driven alignment training, which optimizes for functional equivalence rather than surface-level similarity, enables the memory model to capture sufficient statistics for the agent's decision-making without any online environment interaction. Extensive experiments on SWE-bench Verified across three agent backbones (DeepSWE-32B, Qwen3-Coder-Max, GLM-4.7) demonstrate that COMEM achieves up to $2.08\times$ end-to-end speedup under standard serving and up to $4.95\times$ peak per-step speedup at high concurrency, while retaining competitive resolve rates with the full-context baseline. Furthermore, we show that the framework scales gracefully in deployment: a single 4B memory model can serve approximately 300 concurrent agent instances, rendering the additional architectural complexity negligible at scale.

## Acknowledgements

Our work is sponsored in part by NSF CAREER Award 2239440, NSF Proto-OKN Award 2333790, Sponsored Research Projects from companies like Cisco and eBay, as well as generous gifts from Google, Adobe, and Teradata. Any opinions, findings, and conclusions or recommendations expressed herein are those of the authors and should not be interpreted as necessarily representing the views, either expressed or implied, of the U.S. Government. The U.S. Government is authorized to reproduce and distribute reprints for government purposes not withstanding any copyright annotation hereon.

## Impact Statement

This work primarily advances the efficiency and scalability of Long-Horizon Agentic Systems. By decoupling memory management from reasoning, CoMem significantly reduces the computational resources and energy required to deploy capable autonomous agents. This efficiency contributes to the goal of "Green AI" by lowering the carbon footprint associated with the inference of massive Large Language Models, particularly in memory-bound regimes.

Furthermore, by alleviating the latency bottleneck, this framework lowers the barrier to entry for deploying sophisticated agents in real-time and interactive applications, democratizing access to long-context capabilities on resource-constrained hardware. However, we acknowledge that increasing the efficiency of autonomous agents also lowers the cost of potential malicious use, such as automated cyber-attacks or large-scale social engineering. As the marginal cost of agent deployment decreases, the importance of robust safety alignment and defensive guardrails becomes increasingly critical. We hope our work encourages further research into efficient architectures that prioritize both performance and safety.

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

## A. Prompt for Agent Model

```
Consider the following github issue:
<github_issue>
{problem_statement}
</github_issue>

Can you help me implement the necessary changes to the
repository to fix the <github_issue>? Please refer to the
summary and recent messages for context, which contains our
previous conversations and what you have done so far.

Here are the summary of our previous conversations:
<summary>
{summary}
</summary>
Here are the most recent messages exchanged that are not
included in the summary:
<recent_messages>
{recent_messages}
</recent_messages>
```

## B. Prompt for Memory Model

```
You are a helpful assistant that summarizes conversations
for another model to use. You need to make sure that the
summary captures all important details from the conversation
so that the other model can understand the context just like
reading the full conversation.

### Conversation:
{conversation}

Please provide a detailed summary of the above conversation.
Make sure to include all necessary details for the other
model to understand the context, especially entities, actions
taken, and outcomes. But NEVER include any suggestions or
recommendations for future actions|only summarize what has
already occurred.
```

## C. Additional Baseline Comparison

Our internal baselines are designed to strictly isolate the core contributions of CoMem: the *Full-Context* baseline establishes the performance ceiling without compression, the *No Summary* variant ablates the memory model entirely, and the *Off-the-Shelf* (base) variant evaluates generic summarization without task-specific training. Together, these systematically demonstrate that memory should be decoupled from the agent and trained for downstream decision-making.

To provide an external point of comparison, we additionally evaluate against MemAgent (Yu et al., 2025), a general-purpose long-context compression model that is well suited as a drop-in replacement for our memory component. We integrate the official `BytedTsinghua-SIA/RL-MemoryAgent-7B` checkpoint, keeping all other experimental settings (including the asynchronous $k$-step-off pipeline) identical. We evaluate using both the `GLM-4.7` and `Qwen3-Coder-Max` agent backbones. Results are reported in Table 5.

As shown in Table 5, CoMem consistently outperforms MemAgent across both agent backbones in terms of task effectiveness and inference efficiency. On GLM-4.7, CoMem achieves a 7.9% higher resolve rate while also being faster (2.08× vs.

*Table 5.* Comparison with MemAgent on SWE-bench Verified. All methods use the same asynchronous pipeline and agent backbone. Latency is reported as total inference time for 128 issues (with CPU offload).

| Agent Backbone | Memory Method | %Resolved | Latency (s) | Speedup |
|---|---|---|---|---|
| **GLM-4.7** | Full-Context | 69.0 | 5869.42 | 1.00× |
| | MemAgent (7B) | 54.8 | 3253.83 | 1.80× |
| | **CoMem (GRPO)** | **62.7** | **2821.38** | **2.08×** |
| **Qwen3-Coder-Max** | Full-Context | 57.2 | 5129.90 | 1.00× |
| | MemAgent (7B) | 43.2 | 3672.48 | 1.40× |
| | **CoMem (GRPO)** | **51.0** | **3594.51** | **1.43×** |

1.80× speedup). On Qwen3-Coder-Max, CoMem improves the resolve rate by 7.8% with comparable latency. These results confirm that training the memory model with an action-consistency objective tailored to the downstream agent yields superior compression compared to a general-purpose memory model trained without such alignment.

## D. Generalization to Other Long-Horizon Agent Settings

To evaluate the generalizability of CoMem beyond software engineering tasks, we conduct additional experiments on BrowseComp-EN (Wei et al., 2025), a challenging deep research benchmark that requires multi-step information retrieval over the open web.

The agent operates with three available actions: `google_search`, `scrape`, and `submit_answer`. Each task permits a soft maximum of 30 steps and a hard maximum of 50 steps. We use `GLM-4.7` as the agent LLM with a 128K context window (longer than the SWE setting) and GPT-4.1 as the answer judge. For the CoMem variant, we fine-tune `Qwen3-4B-Instruct-2507` using the same training pipeline described in §3.4, with $k = 1$ at test time. Results are reported in Table 6.

*Table 6.* Results on BrowseComp-EN with `GLM-4.7` as the agent backbone.

| Method | Accuracy (%) |
|---|---|
| Full-Context | 28.1 |
| **CoMem ($k = 1$)** | **32.0** |

CoMem improves accuracy by 3.9% over the full-context baseline, demonstrating that the compressed memory not only reduces context length but can actively improve task performance. We hypothesize that deep research tasks involve large volumes of scraped web content that dilute the agent's attention; a structured summary filters this noise and facilitates more focused decision-making. This result suggests that CoMem generalizes effectively to long-horizon agent settings beyond code generation.

## E. Preservation of Procedural Dependencies in Summaries

A natural concern with compression-based memory is whether long-range procedural dependencies are preserved across many interaction steps. To investigate this, we examine CoMem summaries from `Qwen3-Coder-Max` trajectories with $k = 2$. We observe that the trained memory model learns to discard recoverable noise (e.g., raw file contents, full stack traces) while retaining the procedural details critical for downstream decision-making.

**Case 1: django-17084.** During steps 12–18, the agent discovers that the `Window` class has `contains_over_clause=True` but that this attribute is not checked in `get_aggregation()`. Over 56 subsequent turns of exploration and failed attempts, this diagnostic detail could easily be lost. However, at step 74, the CoMem summary still retains this exact finding, enabling the agent to produce the correct code fix without re-deriving the root cause.

**Case 2: matplotlib-24026.** At steps 27 and 32, the agent identifies the relevant logic in `to_rgba()` and notes a missing import statement. Over 43 subsequent turns involving unrelated exploration, both the algorithmic detail and the missing import are preserved in the summary at step 75, allowing the agent to resolve the bug in a single action.

These examples illustrate that the action-consistency reward (§3.4) incentivizes the memory model to retain precisely those details that are causally relevant to future actions, even when they occur many dozens of steps in the past.

## F. Scheduling Overhead Analysis

A potential concern with the asynchronous $k$-step-off pipeline is whether scheduling and queuing delays in the memory server introduce additional latency visible to the agent. We analyze this from two perspectives.

**Per-step Overhead in Real Trajectories.** From production trajectory data, memory summarization time remains stable at 3.0–4.2s per call throughout the trajectory, with zero effective overhead to the agent since summarization completes well within the $k$-step window during which the agent proceeds independently.

**Queuing Delay under Load.** To further isolate queuing from compute, we benchmark the memory server at varying concurrency levels using the real workload profile (28K input, 1K output tokens). At concurrency 1, time-to-first-token (TTFT) reflects pure compute; the increase at higher concurrency isolates queuing delay. Results are reported in Table 7.

*Table 7.* Memory server performance under varying concurrency. Est. Queue Time is computed as the TTFT increase relative to concurrency 1.

| Conc. | Med. TTFT | Queue Time | TPOT | Throughput |
|---|---|---|---|---|
| 1 | 4,666 ms | 0 ms | 2.9 ms | 3,828 tok/s |
| 8 | 5,678 ms | 1,012 ms | 5.3 ms | 20,845 tok/s |
| 32 | 8,828 ms | 4,162 ms | 14.4 ms | 38,946 tok/s |
| 128 | 23,810 ms | 19,143 ms | 48.5 ms | 46,766 tok/s |

Although concurrency 128 introduces ∼19s of queuing delay, this remains entirely invisible to the agent due to the asynchronous pipeline design: the agent never blocks on memory completion, and the summary is consumed only at the start of the next $k$-step cycle. This confirms that CoMem's scheduling overhead is effectively zero from the agent's perspective, even under heavy memory server load.

## G. Robustness to Scaffold Changes

A practical deployment concern is whether the memory model is brittle to changes in the agent's tool interfaces and prompt templates. To evaluate this, we conduct a cross-scaffold transfer experiment: we take a memory model trained entirely with the R2E-Gym scaffold and `GLM-4.7` as the agent, and evaluate it using the OpenHands scaffold, which differs substantially in tool definitions, output formats, and prompt structure. Despite these differences, the transferred memory model achieves 62.4% resolve rate compared to 62.7% with the original scaffold. This minimal degradation (−0.3%) suggests that the memory model learns to extract and compress task-relevant information in a manner that is largely invariant to the specific tool interfaces and prompt templates used during inference.

## H. Hyperparameters

**Training.** We train the memory model (`Qwen3-4B`) using GRPO with the following hyperparameters. We use a learning rate of $1 \times 10^{-6}$ with a cosine schedule and a warmup ratio of 0.05. The training batch size is 128 prompts with a group size of 16 rollouts per prompt. The clipping parameter is $\epsilon = 0.2$ and the KL penalty coefficient is $\beta = 0.01$. We train for 3 epochs on trajectories collected from R2E-Gym-Lite using the full-context agent pipeline. The maximum input length for the memory model is 32,768 tokens, and the maximum summary output length is capped at 2,048 tokens. For the action-consistency reward, we use cosine similarity between the agent's action given the summary and the ground-truth full-context action. The SFT warmup stage uses the same learning rate and trains for 1 epoch before GRPO fine-tuning.

**Evaluation.** We evaluate on SWE-Bench-Verified using 500 GitHub issues from the test set, processed in batches of 128 issues for latency measurement. The maximum number of agent steps is set to 40, with an extended allowance of up to 100 steps to permit submission. We use the R2E-Gym scaffold with default tool definitions. For the $k$-step-off pipeline, we set $k = 2$ for `DeepSWE` and `Qwen3-Coder-Max`, and $k = 4$ for `GLM-4.7`. The maximum summary length is 2,048 tokens for all configurations. For inference serving, we use vLLM version `0.14.0` with prefix caching and chunked prefilling enabled. `DeepSWE` is served on A100 (80GB) GPUs, and `Qwen3-Coder-Max` and `GLM-4.7` are served on H200 GPUs. The agent model's generation temperature is set to $0.6$ with top-$p$ sampling at $0.95$. The memory model uses greedy decoding (temperature 0) for deterministic summaries.

## I. Cross-Backbone Transferability

We further investigate whether a trained memory model specializes to the particular agent backbone used during training. To test this, we take a memory model trained with `Qwen3-Coder-Max` as the agent backbone and evaluate it directly with `GLM-4.7` without any retraining. The transferred configuration achieves 61.9% resolve rate, compared to 62.7% for the dedicated training with `GLM-4.7`. The modest gap ($-0.8\%$) indicates that the memory model acquires a general-purpose compression strategy that transfers across agent architectures, while dedicated per-backbone training can further elevate performance.

