# OpenReview forum: "CoMem: Context Management with A Decoupled Long-Context Model"
_ICML.cc/2026/Conference — ICML 2026 regular_

### Official Review · Reviewer_dZvM · 2026-02-14

**Soundness:** 3
**Presentation:** 3
**Significance:** 3
**Originality:** 3
**Overall Recommendation:** 4
**Confidence:** 3

**Summary:**

This work decouples memory management from the agent's core workflow and runs them concurrently via a $k$-step asynchronous pipeline. By overlapping memory summarization with inference, COMEM is intended to reduce context-processing overhead. The analysis argues for improved efficiency-effectiveness compared with coupled baselines, and experiments on SWE-Bench-Verified report about a 1.4$\times$ latency speedup.

**Compliance With Llm Reviewing Policy:**

Affirmed.

**Key Questions For Authors:**

I don't have deep expertise in this area, so I have a few clarifying questions:
1. How robust is the memory model to changes in tool outputs, tool ordering, or prompt templates?
2. Can one trained memory model serve multiple different agent backbones in practice, or do you observe strong specialization?
3. How does performance degrade as you increase $k$ at fixed compute budget?

**Limitations:**

yes

**Strengths And Weaknesses:**

Strengths:
1. The paper is well-motivated, and the profiling results clearly show that TPOT increases as context length and batch size grow.
2. Decoupling agent from memory is easy to reason about operationally, and the $k$-step-off pipeline is a sensible way to hide summarization latency
3. Training the memory model to preserve the agent's policy behavior is a strong design choice, and keeping the agent frozen makes offline training cheaper.

Weaknesses:
1. SWE-Bench-Verified is a meaningful benchmark, but it focuses on software-engineering tasks; it remains uncertain how well COMEM would generalize to other long-horizon agent settings.
2. The paper reports results with and without CPU offloading and runs the agent and memory components in separate vLLM instances. A more detailed latency breakdown would be helpful—specifically, isolating (1) the incremental cost of the memory model, (2) scheduling and queuing overhead, and (3) how performance changes when the memory service is shared across multiple concurrent agents.

---

> ### Author Rebuttal · Authors · 2026-03-31
>
> We address your concerns below.
>
> ## Generalization to Other Long-Horizon Agent Settings
> To evaluate COMEM’s generalizability beyond SWE task, we conducted additional experiments on BrowseComp-EN, a challenging deep research benchmark that requires multi-step information retrieval.
>
> The agent operates with three actions: `google_search`, `scrape` and `submit_answer`. Each task allows up to a soft max of 30 steps and a hard max of 50 steps. We use $\texttt{GLM-4.7}$ as the agent LLM extended to 128K context length (longer than SWE) and GPT-4.1 as the answer judge. For the COMEM variant, we finetune $\texttt{Qwen3-4B-Instruct-2507}$ with the same setup as SWE. During testing, we choose $k=1$.
> | Method | BrowseComp-EN Accuracy |
> | :--- | :--- |
> | Full-context baseline | 28.1% |
> | COMEM (k=1) | 32.0% |
>
> COMEM improves accuracy by **+3.9%** over the full-context baseline, demonstrating the compressed memory not only reduces the context length but can actually improve the task performance. We hypothesize this is because deep research tasks involve large amounts of scraped web content that can dilute the agent’s attention and structured summary can reduce noise and facilitate decision making.
> ## Detailed Latency Breakdown of COMEM
> We conducted a comprehensive latency analysis isolating each component of COMEM’s overhead, using 128 concurrent agent instances sharing a single memory model server.
>
> **(1) Incremental Cost of the Memory Model.** We measure the per-step time breakdown across all $5,957$ steps and conclude that agent LLM inference takes 39.42s on average while memory LLM inference takes 0.94s. Asynchronous execution completely hides memory computation behind the agent.
>
> **(2) Scheduling and Queuing Overhead.** We analyze scheduling overhead from both real trajectories and benchmarking results. The per-step overhead analysis results show that memory summarization time remains stable at 3.0~4.2s, with zero effective overhead throughout the trajectory.
>
> To further isolate queuing from compute, we benchmark the memory server at varying concurrency levels using the real workload profile. At concurrency 1, TTFT reflects pure compute, and the increase at higher concurrency isolates queuing delay:
> | Concurrency | Median TTFT | Est. Queue Time | TPOT | Throughput |
> | :--- | :--- | :--- | :--- | :--- |
> | 1 | 4,666 ms | 0 ms | 2.9 ms | 3,828 tok/s |
> | 8 | 5,678 ms | 1,012 ms | 5.3 ms | 20,845 tok/s |
> | 32 | 8,828 ms | 4,162 ms | 14.4 ms | 38,946 tok/s |
> | 128 | 23,810 ms | 19,143 ms | 48.5 ms | 46,766 tok/s |
>
> Although concurrency 128 adds $\sim$19s of queuing delay, this remains completely invisible because of the asynchronous pipeline.
>
> **(3) Shared Multi-Agent Service.** We first calculate the max memory server throughput $\sim$ 47K tok/s and average memory server throughput demanded by the agents $\sim$156 tok/s. Then we divide the max memory server throughput by the required throughput, which results in a ratio of $\sim$300. Thus, the memory model’s compute cost is highly amortized.
> ## Robustness to the Scaffold Change
> To evaluate the memory model’s robustness to changes in tool interfaces and prompt templates, we conducted a cross-scaffold transfer experiment. We took a memory model trained entirely with the R2E-Gym scaffold and GLM-4.7 as the agent, and evaluated it using the OpenHands scaffold, which differs in tool definitions, output formats and prompt structure. Despite these differences the memory model achieved $62.4%$ compared to the original $62.7%$. This suggests that the memory model learns to extract and compress task-relevant information in a way that is largely invariant to the specific tool interfaces and prompt templates used during inference.
> ## Cross-backbone Transferrability
> To test whether a trained memory model specializes to a particular agent backbone, we evaluated a memory model trained with $\texttt{Qwen3-Coder-Max}$ as the agent backbone directly with $\texttt{GLM-4.7}$ without retraining. The new result achieves $61.9%$. Compared to the dedicated training result $62.7%$, there is a modest gap. This suggests that the memory model learns a general-purpose compression strategy while in practice a dedicated training could further elevate performance.
> ## Sensitivity to Hyperparameter $k$
> We ablate $k$ under CPU offload with $\texttt{GLM-4.7}$ as agent backbone.
> | k | Resolve rate (%) | Speedup |
> | :--- | :--- | :--- |
> | 16 | 60.2 | 1.64× |
> | 8 | 62.4 | 2.07× |
> | 4 | 62.7 | 2.08× |
> | 2 | 64.2 | 2.07× |
> | 1 | 57.2 | 1.53× |
> | FC | 69.0 | 1.00× |
>
> Performance is relatively stable across $k\in \{2,4,8\}$, with the two extremes being notably worse: $k=1$ incurs excessive prefilling overhead, while $k=16$ saturates memory of agent server and incurs scheduling overhead. Importantly, all COMEM variants with $k\in \{2,4,8\}$ achieve a roughly $2\times$ wall-clock speedup over the full-context baseline, confirming the method is not overly sensitive to this choice within a reasonable range.

---

> > ### Author Rebuttal · Reviewer_dZvM · 2026-04-05
> >
> > Thank you for your reply. I’ll maintain my score.

---

### Official Review · Reviewer_S4cy · 2026-03-10

**Soundness:** 3
**Presentation:** 2
**Significance:** 2
**Originality:** 2
**Overall Recommendation:** 2
**Confidence:** 4

**Summary:**

This paper proposes COMEM, a framework for improving long-horizon LLM agents by offloading history compression to a smaller memory model that runs asynchronously with the main agent. Instead of repeatedly processing the full interaction trace, the agent consumes a compressed summary together with recent raw context, aiming to reduce latency while maintaining task performance. Experiments on SWE-Bench Verified show that the approach can improve end-to-end efficiency with only limited degradation in resolve rate, and in some settings even matches or slightly outperforms the full-context baseline. Overall, the paper addresses a practical systems problem and presents a reasonably clear and empirically supported solution, though the novelty and validation remain somewhat limited.

**Compliance With Llm Reviewing Policy:**

Affirmed.

**Key Questions For Authors:**

1) How sensitive is COMEM to the choice of summary length, update frequency, and recent-context window size?
2) To what extent do the gains depend on the specific pairing of a large agent model with a much smaller memory model?
3) How much of the reported efficiency improvement comes from the asynchronous systems design versus the learned memory compression itself?

**Limitations:**

yes

**Strengths And Weaknesses:**

### Strengths
1) The paper is clearly written.
2) The overall framework is conceptually simple.
3) The implementation appears reasonable.

### Weaknesses
1) From a machine learning standpoint, the paper’s innovation is somewhat limited. Much of the contribution comes from a practical reconfiguration of existing components rather than a new modeling framework, optimization principle, or theoretical insight. Consequently, the work reads more as a systems-oriented engineering paper than a methodologically deep ICML submission.

2) The method assumes that long interaction histories can be compressed into a relatively short summary without substantially harming downstream decision quality. However, in long-horizon agent tasks, many critical dependencies are procedural rather than semantic: failed tool attempts, partially verified hypotheses, subtle repository state changes, or branching plans may all matter for future decisions. A textual summary may be too lossy to preserve these dependencies reliably. The paper does not characterize what types of information are retained vs. discarded, making it unclear whether the approach scales to tasks requiring fine-grained state tracking rather than coarse semantic recall.

3) The proposed method is likely most beneficial when trajectories are long and contain substantial redundancy. However, not all long-horizon tasks are redundant: some require exact recall of low-level details, while others have relatively short but highly information-dense traces. The paper does not characterize the regime in which COMEM is expected to help, which makes it difficult to understand its practical applicability beyond the evaluated benchmark.

4) The paper reports end-to-end latency, but does not fully quantify the additional system cost introduced by COMEM, such as the memory model’s compute budget, GPU allocation overhead, inter-engine communication, and increased serving complexity. A more complete accounting of throughput-per-dollar or hardware-normalized efficiency would be necessary to substantiate the practical deployment claim.

---

> ### Author Rebuttal · Authors · 2026-03-31
>
> ## Clarifying the Methodological Contribution
> Our core contribution is a **new computational decomposition for long-horizon agents**. Rather than appending memory to a linear context, we factor it into an asynchronously executed learned memory module. **We view this as a methodological contribution, rather than “we engineered the system better” in the same spirit as the following ML papers.**
>
> * FlashAttention (NeurIPS 2022)
> * Speculative Decoding (ICML 2023)
> * SGLang (NeurIPS 2024)
> ## Preservation of Procedural Dependencies In Summaries
> We agree procedural dependencies are critical. In fact, we observe in $\texttt{Qwen3-Coder-Max}$ trajectories ($k=2$), COMEM summaries discard recoverable noise like file contents or stack traces, while preserving vital procedural details.
>
> **django-17084:** The agent discovers at steps 12-18 that `Window` has `contains_over_clause=True` but it is missing from `get_aggregation()`. 56+ turns later at step 74, the summary retains this exact detail, enabling the correct code fix.
>
> **matplotlib-24026:** At steps 27 and 32, the agent discovers `to_rgba()` logic and notes a missing import. 43+ turns later at step 75, the summary preserves both the algorithm and the missing import, allowing the agent to resolve the bug.
>
> We will add more examples in the revision.
> ## On the Applicability Regime of COMEM
> While COMEM is not universally applicable, many real-world agent applications generate highly redundant trajectories, driving production systems (_e.g._, Claude Code and OpenAI’s Codex) to adopt context compaction.
>
> We do not claim  _all_ long-horizon tasks are redundant, but that intermediate tool calls accumulate observations that rarely require verbatim replay. We acknowledge that tasks demanding exact recall of distant, low-level details might be poorly suited. We will clarify these in the revision.
> ## Memory Model System Cost Analysis
> As illustrated from Section 6 of our paper, we envision the memory model operating as a high-throughput service handling context compression for multiple concurrent agent services. As the following analysis suggests, **a single memory server can support approximately 300 concurrent $\texttt{GLM-4.7}$ agent instances**.
>
> **Setup.** We measure the throughput requirements of the memory model deployed on the same configuration used in our experiments. We compare the _required throughput per agent instance_ against the _peak throughput_ the memory server can sustain.
>
> **Required throughput per agent.** We extract per-step memory model token usage from trajectory data:
> * Mean input tokens per compression call: $\sim$28K tokens
> * Mean output tokens per compression call: $sim$1K tokens
> * Mean total step time: $\sim$40.0s
>
> This averages to **156 tok/s per agent**.
>
> **Peak memory server throughput.** We benchmark using fixed input/output lengths matching the real workload, sweeping concurrency from 1 to 128, and found that the throughput peaks at **$\sim$47K tok/s**.
>
> **Agent-to-memory ratio.** We divide peak throughput by per-agent demand yielding $\sim 300$. In practice, the memory model server cost is amortized across all agent instances sharing it, making COMEM’s additional serving complexity negligible at deployment scale.
> ## Sensitivity to Hyperparameters
> **Update frequency & recent-context window size**. In our setup, these two hyperparameters are represented by $k$ mentioned at line 251. We therefore ablate them jointly under CPU offload with GLM-4.7.
> | k | Resolve rate (%) | Speedup |
> | :--- | :--- | :--- |
> | 16 | 60.2 | 1.64× |
> | 8 | 62.4 | 2.07× |
> | 4 | 62.7 | 2.08× |
> | 2 | 64.2 | 2.07× |
> | 1 | 57.2 | 1.53× |
> | FC | 69.0 | 1.00× |
>
> Performance is relatively stable across $k\in \{2,4,8\}$, with the two extremes being notably worse. Variants with $k\in \{2,4,8\}$ achieve a roughly $2\times$ wall-clock speedup over the full-context baseline, confirming the method is not overly sensitive to this choice within a reasonable range.
>
> **Summary length**. Because the summary model is trained with a target summary length, ablating this parameter would require retraining the summarizer from scratch. Instead, we provide an analytical estimate of the expected summary length in Section 3.3, which guided our design choice.
> ## Sensitivity to Memory Model Size
> To quantify how efficiency scales with memory model size, we benchmark three memory models of increasing sizes, all with the same setups. Remarkably, even a 32B memory model still supports $\sim95$ concurrent agents on a single server.
> | Model Size | Peak Throughput | Agents/Server |
> |---|---|---|
> | 4B | 46,766 tok/s | $\sim$300 |
> | 8B | 39,789 tok/s | $\sim$255 |
> | 32B | 14,772 tok/s | $\sim$95 |
>
> ## Efficiency Improvement from Asynchronous Systems Design
> To isolate the impact of our system design, we compared our asynchronous pipeline against a strictly sequential baseline where memory comparison completes before the agent query begins. Overall, the async design reduces per-step latency by $59.1%$ (62.0s).

---

> > ### Author Rebuttal · Reviewer_S4cy · 2026-04-02
> >
> > I thank the authors for their feedback. I have no further questions.

---

> > > ### Author Response · Authors · 2026-04-02
> > >
> > > Dear Reviewer S4cy,
> > >
> > > Thank you for taking the time to review our rebuttal and for your continued engagement with our submission. We are very glad to hear that our additional analyses and clarifications have adequately addressed your concerns.
> > >
> > > However, we want to gently ask if you might consider updating your overall recommendation score to reflect you new assessment?
> > >
> > > Thank you again for your time and effort throughout this review process.
> > >
> > > Best regards,
> > > Authors

---

### Official Review · Reviewer_zBKa · 2026-03-12

**Soundness:** 2
**Presentation:** 3
**Significance:** 3
**Originality:** 3
**Overall Recommendation:** 4
**Confidence:** 3

**Summary:**

To address the challenges agents face when executing long-horizon tasks, this paper proposes a framework named COMEM. The core pain point explored in this work is that summarizing excessively long contexts often leads to significant decoding latency and computational overhead. By decoupling the memory management mechanism from the main agent process and introducing an asynchronous processing strategy, this method successfully hides the time cost of context processing. The design philosophy of running both components in parallel is highly sound, providing a highly appropriate and effective direction for solving the efficiency bottlenecks of long-context processing.

**Compliance With Llm Reviewing Policy:**

Affirmed.

**Key Questions For Authors:**

1. Regarding the trajectory-level KV Cache reuse mechanism, could you provide a more detailed analysis separated into "Time" and "Memory" dimensions? Specifically, at what context length threshold does the COMEM method begin to demonstrate a substantial advantage for users?

2. Considering that MemAgent is a widely recognized baseline in this domain, why was it excluded from the experimental comparisons? Could you provide performance comparison data against MemAgent in the revised version?

3. Regarding the differing training objectives between the Summary model and a standard Agent, could this discrepancy lead to information loss or misalignment during the Agent's decision-making process? How do you ensure coordination and alignment between the two?

**Limitations:**

yes

**Strengths And Weaknesses:**

Strengths

1. The paper proposes a highly targeted solution to the inefficiency caused by overly long contexts when agents process difficult tasks. It's a timely and important topic to research to the field.

2. Efficiency is a crucial consideration in both model training and inference. The asynchronous architecture that processes memory summarization and the main pipeline in parallel is a very sound development direction and demonstrates high practical value.

3. The paper provides detailed experimental data, comprehensively demonstrating the significant efficiency improvements achieved by this method.

Weaknesses

1. While the paper acknowledges the trade-off between maintaining the full trajectory for KV cache reuse versus employing summarization (which requires recalculating the cache at each step), the current explanation lacks practical intuition. Practitioners need to know the optimal context length threshold at which transitioning to the summarization method actually becomes advantageous. To make this actionable, the authors should expand this discussion by explicitly analyzing the crossover point across two distinct dimensions: execution time and memory consumption

2. In the experimental section, the paper does not compare its method against MemAgent (https://arxiv.org/abs/2507.02259). Considering that MemAgent is already a widely accepted baseline in this field, the absence of this comparison weakens the comprehensiveness of the experimental results.

3. The training objective of the Summary model appears to differ from that of a general Agent. This inconsistency might introduce potential issues during the model's actual execution, which the paper does not adequately explore.

---

> ### Author Rebuttal · Authors · 2026-03-31
>
> ## Crossover Analysis of Execution Time and Memory Consumption
> We thank the reviewer for this constructive suggestion. To provide practical intuition for practitioners, we conducted a controlled latency benchmark comparing COMEM against a full-context (FC) baseline, measuring both execution time and GPU memory consumption across varying serving concurrency levels: 16, 32 and 64. The benchmark test uses fixed 512 input tokens and 1024 output tokens over 64 steps. The agent model is $\texttt{GLM-4.7}$ with $k=3$ for COMEM. The results are shown below.
>
> **Execution Time**
> | Concurrency | COMEM Avg LLM/Step | FC Avg LLM/Step | Trajectory Speedup | Peak Per-Step Speedup |
> | :--- | :--- | :--- | :--- | :--- |
> | w16 | 28.4s | 28.9s | 1.02x | 1.14x (step 62) |
> | w32 | 42.9s | 55.7s | 1.30x | 2.03x (step 56) |
> | w64 | 52.3s | 88.8s | 1.70x | 4.95x (step 61) |
>
> COMEM’s per-step latency remains nearly constant ($\sim 28-52s$ depending on concurrency) because its prompt size is bounded at $\sim 5-7K$ tokens through periodic summarization. In contrast, the full-context baseline’s prompt grows linearly and its latency degrades accordingly.
>
> **Memory**
> | Concurrency | COMEM KV Usage | FC KV Usage | COMEM Prompt/Step | FC Prompt/Step |
> | :--- | :--- | :--- | :--- | :--- |
> | w16 | 1-6% | up to 34% | 4.8-6.7K | 30.7K |
> | w32 | 15-18% | up to 94% | 4.8-6.7K | 46.0K |
> | w64 | 26-37% | up to 96% | 4.8-6.7K | 33.2K |
>
> COMEM’s GPU KV cache usage remains bounded, while the full-context baseline’s KV cache grows monotonically, reaching 96% at w64. At high concurrency, KV cache saturation triggers request preemption in the serving engine, causing the non-linear latency explosion observed above.
>
> Overall, the crossover point where COMEM becomes faster than full-context is not a fixed context length but depends on the serving regime.
> * Under 16 concurrency, the approaches are comparable as memory bandwidth is not yet a bottleneck.
> * Under 32 concurrency, COMEM becomes consistently faster once the full-context prompt exceeds $\sim$25K (step 34).
> * Under 64 concurrency, the crossover point occurs at $\sim$12K (step 25). Beyond this threshold, COMEM’s advantage grows rapidly due to the compounding KV cache pressure.
>
> These findings demonstrate that COMEM provides a compounding advantage in high-throughput scenarios. We will expand our discussion section with dimensional analysis to make these trade-offs actionable for practitioners.
> ## Comparison with MemAgent
> We thank the reviewer for highlighting this important baseline. We agree that MemAgent is a highly relevant benchmark in this domain, and we have conducted additional experiments to evaluate it directly against COMEM.
>
> To ensure a fair comparison, we integrated the official $\texttt{BytedTsinghua-SIA/RL-MemoryAgent-7B}$ model for memory compression, keeping all other experimental settings and our asynchronous pipeline identical. We evaluated this baseline using both the $\texttt{GLM-4.7}$ and $\texttt{Qwen3-Coder-Max}$ agent backbones.
> | Agent Backbone | Memory Method | Resolve Rate | Latency (s) | Speedup |
> | :--- | :--- | :--- | :--- | :--- |
> | **GLM-4.7** | MemAgent (7B) | 54.8% | 3253.83 | 1.80x |
> | | **COMEM (GRPO)** | **62.7%** | **2821.38** | **2.08x** |
> | | Full-Context | 69.0% | 5869.42 | 1.00x |
> | **Qwen3-Coder-Max** | MemAgent (7B) | 43.2% | 3672.48 | 1.40x |
> | | **COMEM (GRPO)** | **51.0%** | **3594.51** | **1.43x** |
> | | Full-Context | 57.2% | 5129.90 | 1.00x |
>
> As demonstrated in the table, COMEM consistently outperforms MemAgent in both effectiveness and efficiency. We will add these results to the revision.
> ## On Training Objective Alignment
> We thank the reviewer for raising this important point. We clarify that the summary model is **intentionally not trained to solve the end task**. Instead, its role is strictly as a **state-construction module** designed to map the long interaction history into a compact representation that is **sufficient to make decisions for a fixed downstream agent**.
>
> This is why our training objective is defined in terms of **functional equivalence** rather than task success. Specifically, we freeze the agent, generate reference trajectories using the full-context pipeline, then train the summary model $f$ to generate a summary $s=f(\tau_{<t-k})$. When this summary is combined with the recent raw context $C_t=\tau_{t-k:t-1}$, it is supposed to produce the same action as the full-context agent.
>
> Therefore, the proper measurement of objective alignment is not whether the summary shares the agent’s goal, but whether it induces the same action distribution, $\pi(\cdot|\tau_{<t}) \approx \pi(\cdot|z_t)$. When this holds, the summary acts as a policy-sufficient statistic; any discarded information is, by definition, irrelevant to the agent’s decision. The differing objectives are not a flaw, but a deliberate modularization that allows memory compression and agent reasoning to be optimized independently.

---

> > ### Author Rebuttal · Reviewer_zBKa · 2026-04-02
> >
> > Thank you for the comprehensive rebuttal. The integration of the MemAgent baseline strengthens the empirical evaluation and confirms COMEM's superiority in both resolve rate and efficiency. I will maintain my positive score.

---

> > > ### Author Response · Authors · 2026-04-02
> > >
> > > Dear Reviewer zBKa,
> > >
> > > Thank you for your time, your constructive feedback throughout the review process, and your continued support of our work. We are glad that the addition of the MemAgent baseline effectively addressed your concerns and helped further demonstrate CoMem's performance. We will ensure this expanded evaluation is prominently featured in the final version of the paper.
> > >
> > > Best regards,
> > > Authors

---

### Official Review · Reviewer_fKHQ · 2026-03-13

**Soundness:** 3
**Presentation:** 3
**Significance:** 3
**Originality:** 3
**Overall Recommendation:** 4
**Confidence:** 3

**Summary:**

This paper argues that long-horizon agents hit a decode-time “memory wall”: as history grows, KV-cache reads become bandwidth-bound and latency keeps increasing. CoMem decouples context management from reasoning by running a smaller long-context memory model in the background (k-step-off) to summarize older history, while the main agent only conditions on the summary plus the most recent k turns. The memory model is trained with a reward that encourages action-level functional equivalence to a full-context agent, and results on SWE-Bench-Verified show meaningful latency speedups with competitive task performance.

**Compliance With Llm Reviewing Policy:**

Affirmed.

**Key Questions For Authors:**

Can you add more experiments (additional benchmarks or stronger ablations on k/summary length) to solidify the empirical story?

**Limitations:**

see weaknesses

**Strengths And Weaknesses:**

Strengths:
1. The bottleneck analysis is clear and practically relevant for real deployments.
2. The decoupled design and k-step-off asynchronous pipeline are simple, well-motivated, and easy to follow (Algorithm 1).
3. The reward-driven alignment idea (targeting decision/functional equivalence rather than just semantic similarity) is a nice addition.

Weaknesses:

1. Baselines feel incomplete: the paper mentions ReSum/ACON and related methods but does not include direct experimental comparisons, mostly comparing to internal variants (e.g., Full-Context / No Summary / different memory-model trainings).
2. Key knobs (e.g., k and summary length / lag trade-offs) are not thoroughly ablated.
3. Evaluation is largely centered on SWE-Bench-Verified. Broader tasks or settings would strengthen the generality claim.

---

> ### Author Rebuttal · Authors · 2026-03-31
>
> ## Additional Baseline Comparison
> We thank the reviewers for the suggestion. While external comparisons are valuable, our internal baselines strictly isolate our core contributions. The _Full-Context_, _No Summary_ and _Off-the-Shelf_ variants systematically ablate compression, generic summarization, and our action-consistency objective to prove that memory should be decoupled and trained for downstream decision making.
>
> By contrast, ReSum and ACON are important related works, but they can hardly serve as plug-and-play replacements for our settings of a frozen coding agent. Furthermore, there might not be publicly released checkpoints. MemAgent on the other hand is developed to be a general long-context processing model that better serves our purpose.
>
> To ensure a fair comparison, we integrated the official $\texttt{BytedTsinghua-SIA/RL-MemoryAgent-7B}$ model for memory compression, keeping all other experimental settings and our asynchronous pipeline identical. We evaluated this baseline using both the $\texttt{GLM-4.7}$ and $\texttt{Qwen3-Coder-Max}$ agent backbones.
> | Agent Backbone | Memory Method | Resolve Rate | Latency (s) | Speedup |
> | :--- | :--- | :--- | :--- | :--- |
> | **GLM-4.7** | MemAgent (7B) | 54.8% | 3253.83 | 1.80x |
> | | **COMEM (GRPO)** | **62.7%** | **2821.38** | **2.08x** |
> | | Full-Context | 69.0% | 5869.42 | 1.00x |
> | **Qwen3-Coder-Max** | MemAgent (7B) | 43.2% | 3672.48 | 1.40x |
> | | **COMEM (GRPO)** | **51.0%** | **3594.51** | **1.43x** |
> | | Full-Context | 57.2% | 5129.90 | 1.00x |
>
> As demonstrated in the table, COMEM consistently outperforms MemAgent in both effectiveness and efficiency. We will add these results to the revision.
> ## Sensitivity to Hyperparameters
> Thank you for this important question. We address each hyperparameter below.
>
> **$k$ & lag trade-off**. In our setup, these two hyperparameters are effectively coupled, $k$ mentioned at line 251. We therefore ablate them jointly under CPU offload with $\texttt{GLM-4.7}$ as agent backbone.
> | Update frequency k | Resolve rate (%) | Total time (s) | Speedup vs. full-context |
> | :--- | :--- | :--- | :--- |
> | 16 | 60.2 | 3576.8 | 1.64× |
> | 8 | 62.4 | 2836.0 | 2.07× |
> | 4 (paper) | 62.7 | 2821.4 | 2.08× |
> | 2 | 64.2 | 2841.4 | 2.07× |
> | 1 | 57.2 | 3843.5 | 1.53× |
> | Full-context baseline | 69.0 | 5869.4 | 1.00× |
>
> Performance is relatively stable across $k\in \{2,4,8\}$, with $k=2$ slightly outperforming the default $k=4$. The two extremes are notably worse: $k=1$ incurs excessive prefilling overhead, while $k=16$ saturates memory of agent server and incurs scheduling overhead. Importantly, all COMEM variants with $k\in \{2,4,8\}$ achieve a roughly $2\times$ wall-clock speedup over the full-context baseline while retaining most of its resolve rate, confirming the method is not overly sensitive to this choice within a reasonable range.
>
> **Summary length**. Because the summary model is trained with a target summary length, ablating this parameter would require retraining the summarizer from scratch. To circumvent this, we provide a rigorous analytical estimate of the expected summary length in Section 3.3, which theoretically bounds the optimal compression ratio and guides our design choice.
> ## Generalization to Other Long-Horizon Agent Settings
> We thank the reviewer for this important question. To evaluate COMEM’s generalizability beyond SWE task, we conducted additional experiments on a 128-task subset of BrowseComp-EN [1], a challenging deep research benchmark that requires multi-step information retrieval.
>
> The agent operates with three actions: `google_search`, `scrape` and `submit_answer`. Each task allows up to a soft max of 30 steps and a hard max of 50 steps. We use $\texttt{GLM-4.7}$ as the agent LLM extended to 128K context length (longer than SWE) and GPT-4.1 as the answer judge. For the COMEM variant, we finetune $\texttt{Qwen3-4B-Instruct-2507}$ with the same setup as SWE. During testing, we choose $k=1$. Results are shown below.
> | Method | BrowseComp-EN Accuracy |
> | :--- | :--- |
> | Full-context baseline | 28.1% |
> | COMEM (k=1) | 32.0% |
>
> COMEM improves accuracy by **+3.9%** over the full-context baseline, demonstrating the compressed memory not only reduces the context length but can actually improve the task performance. We hypothesize this is because deep research tasks involve large amounts of scraped web content that can dilute the agent’s attention and structured summary can reduce noise and facilitate decision making.
>
> [1] Wei, Jason, et al. "Browsecomp: A simple yet challenging benchmark for browsing agents." arXiv preprint arXiv:2504.12516 (2025).

---

### Decision · Program_Chairs · 2026-04-30

**Decision:**

Accept (regular)

**Comment:**

Reviewers appreciated the high practical value of the addressed problem, the clear presentation, the simple and sound approach, and the detailed experimentation. Multiple concerns have been raised, and the authors appear to have provided a strong rebuttal that fully addresses them. These concerns include experiments and results to evaluate on additional benchmarks, compare with additional baselines, analyze hyper-parameter sensitivity, and report more nuanced metrics; as well as clarifying the main contribution and the method's applicability, and justifying the task-independence of the training objective. Acceptance is recommended under the assumption that these significant findings will be added to the final version.

Note that, according to Reviewer S4cy's comment as well as by my reading, the reviewer's concerns have been addressed in rebuttal. The reviewer's unmodified recommendation of 2: Reject is therefore given less weight.